# Patterns of Racing and Career Duration of Racing Greyhounds in New Zealand

**DOI:** 10.3390/ani10050796

**Published:** 2020-05-05

**Authors:** Anna L. Palmer, Charlotte F. Bolwell, Kevin J. Stafford, Arnon Gal, Chris W. Rogers

**Affiliations:** 1School of Veterinary Science, Massey University, Private Bag 11-222, 4442 Palmerston North, New Zealand; c.bolwell@massey.ac.nz (C.F.B.); c.w.rogers@massey.ac.nz (C.W.R.); 2School of Agriculture and Environment, Massey University, Private Bag 11-222, 4442 Palmerston North, New Zealand; k.j.stafford@massey.ac.nz; 3Department of Veterinary Clinical Medicine, College of Veterinary Medicine, University of Illinois at Urbana-Champaign, 1008 W Hazelwood Drive, IL 61802, USA; agal2@illinois.edu

**Keywords:** greyhound, greyhound racing, career length, racing pattern

## Abstract

**Simple Summary:**

There is limited information on the career length and patterns of racing for greyhounds. Performance outcomes, including number of racing starts, career length and the age at which greyhounds finish racing, provide insight into causes of attrition in the greyhound racing industry. To investigate trends in greyhound racing careers a baseline is required. This paper presents results from a retrospective cohort study exploring career duration and patterns of racing of greyhounds in New Zealand.

**Abstract:**

The welfare and wastage of racing greyhounds is a topic of public concern. Little is published about the racing patterns of these dogs in New Zealand. The aim of this study is to describe the pattern of greyhound racing in New Zealand. Data on all race starts between 1 August 2011 and 25 March 2018 were supplied by Greyhound Racing New Zealand. A cohort was created containing dogs that had a racing career between 1 August 2013 and 31 July 2017. Data were collated within a customized Microsoft Access database from electronic records of all racing starts for every dog within the 2013–2016 racing seasons. For this cohort of racing dogs, there were 97,973 race starts across 22,277 races involving 2393 individual greyhounds. The median number of days between racing starts was 7 days (inter-quartile range (IQR): 4–10 days). The median career length was 424 days (IQR: 206–647 days) and the median number of racing starts throughout a racing career was 35 (IQR: 16–59 starts). Dogs of similar ability finished their career at a similar age.

## 1. Introduction

In Australasia, there is public concern about the welfare of racing greyhounds (*Canis familiaris*) with particular focus on injury rates and the “wastage” of greyhounds in the industry [1,2]. Such concerns criticise the greyhound racing industry for exploiting the natural instincts of these sighthounds, in order to create a product, through gambling on a competition based on physical performance and speed [3]. Previous studies have reported descriptions of racing related injuries that occur in racing greyhounds [4,5,6]. However, no studies have investigated patterns of racing and the impact of these on the racing careers of greyhounds; an essential step in recognising areas of potential welfare compromise.

Greyhound racing is one of the three animal racing codes in New Zealand, the other two being Thoroughbred and Standardbred horse racing. It has the lowest number of racing animals and annual financial turnover [7]. The Greyhound industry is estimated to contribute $92.6 million annually to the New Zealand economy [7]. The availability of racing opportunities determines wagering turnover, the primary income stream in the industry [8]. However, racing opportunities may be impacted by the limits of a dog’s physiological ability to maintain regular high-intensity racing and training regimes. 

As part of normal race-day procedures, greyhounds travel to the racetrack and compete in fields against dogs with similar ability. The New Zealand greyhound racing scene is comprised of dogs that were born and registered in New Zealand, as well as dogs that were born and registered in Australia before being registered and raced in New Zealand. These two populations provide an opportunity to examine the impact of the pattern of New Zealand racing on dogs that may have previously raced within another jurisdiction, namely Australia. 

The majority of research into the racing industries, looking in particular at career length, wastage and injuries, has focused on horses, especially thoroughbreds. In thoroughbred racing, the pattern of racing and race training have been associated with increased risk for different injuries [9,10]. Performance outcomes used to quantify wastage in racehorses have included career length and number of lifetime racing starts [11,12,13,14,15]. Injuries sustained by racing greyhounds may compromise an individual’s welfare and pose a direct cost on the Greyhound racing industry through lost racing days, expenses involved with treatments and losses due to early retirement from racing [6,16]. There is no published research surrounding the management, training and racing of greyhounds in New Zealand. However, a few studies have described race level parameters while reporting descriptions of injuries and injury rates [5,6,16]. Quantification of the current industry in New Zealand is necessary in order to provide insight into the potential causes of attrition of racing greyhounds in New Zealand.

The aim of this paper was to describe the racing careers of greyhounds in New Zealand. We present career duration and pattern of racing data that will help identify potential factors that could shorten a dogs racing career or lead to wastage in the industry. 

## 2. Materials and Methods

Greyhound Racing New Zealand (GRNZ), the governing body for greyhound racing in New Zealand holds all the official records of the industry. They provided data containing details on all racing starts in New Zealand between 1 August 2011 and 25 March 2018. These data were used for a retrospective cohort study to investigate all greyhound racing starts from the seven greyhound racing tracks across New Zealand during four racing seasons. The original dataset contained race date, racetrack, race number, race distance, race grade, field size and final placing. The race grade refers to the grade in which the dog was racing, where class 0 (C0) are maiden races, class 1 (C1) are lower grade races and there is a sequential increase to the higher-grade races until class 5 (C5). Dog level data were provided in a separate dataset, which contained information from all dogs registered for racing that were born between 1 August 2009 to 30 November 2017. The data included: Dog identity number, dog name, birth date, dog sex, registration date, dog ear-brand tattoo, and first trial date. These data were imported into a customised Microsoft Access database. Using the race date and dog name information, new variables were created for race season (winter: June–August, spring: September–November, summer: December–February, autumn: March–May), race year, number of racing starts per career, length of career in days, and number of days since previous race for each dog. The dog ear-brand tattoo code was used to create a new variable for the country, New Zealand or Australia, in which the dog was born. The integrity of the data were checked using histograms and cross tabulation, where outliers or points of interest were compared with the official online GRNZ database. 

For this study a cohort containing information on all dogs that started racing after 1 January 2013 and finished racing by 31 July 2017 was created. One of the selection criteria for eligibility was dogs had to be born after 1 July 2011. This date was 18 months before the first race date (1 January 2013) and at this time dogs were not permitted to be nominated for racing until they reach 14 months of age (Rule 19.10) [17]. Data on all racing starts were provided through to 25 March 2018 and data from dogs with no racing records after 31 July 2017 were censored as they were assumed to have finished their racing career.

Descriptive statistics were used to describe the data at a population and at a cohort level and were carried out for the both dog-level and race-level variables. Given the potential differences in the cohort of dogs previously registered and raced in Australia, analyses were stratified by country of birth. Racing milestones of interest included: age when first registered for racing, age at qualifying trial, age of first race start and age of the last recorded racing start. Normality of continuous data were assessed with the Shapiro–Wilk test and Pearson’s chi-squared tests. Continuous data that were non-normally distributed were summarised with medians and percentiles. Nominal data are presented as counts and percentages. Kruskal–Wallis tests and Dunn’s Multiple Comparisons tests were used to determine differences in patterns of racing for race level variables. Kruskal–Wallis tests and Wilcoxson rank-sum tests were used to compare the age at which racing milestones were achieved by the country in which the dog was born. 

The stability of frequency of racing was established using an autocorrelation function applied to data recording the pattern of racing. The distribution of racing starts was explored by categorising the continuous variable of number of days between racing starts for each dog by quartiles thereby creating a categorical variable which contained “high-intensity racing interval” (minimum to lower quartile: 1–4 days), “medium-intensity racing interval” (inter-quartile range: 5–10 days) and “low-intensity racing interval” (upper quartile to maximum: 11–539 days) dogs. The proportion of high-intensity racing intervals that occurred during a racing career was calculated for each dog, using the number of high-intensity racing intervals and the total number of racing starts. Length of racing career was measured as both the number of racing starts during a career and the number of days from the dog’s first racing start through to the last recorded racing start. A Kaplan–Meier curve was applied to quantify the median age of final race start. Differences in the median age at which dogs finished racing were investigated using a log-rank test. Cox regression analysis, using the Breslow method of handling ties, assessed the association between the proportions of high inter-race intervals (adjusting for country of origin, sex of the dog, maximum race grade reached during career) and month started racing, on the outcome of final race age in months. Variables showing some univariate association (*p* < 0.2) with the outcome were further evaluated in a multivariable model with each variable being sequentially removed from the full model in a backward stepwise fashion. However, given that the variables were all inter-related, the multivariable model was not further analysed. All statistical analyses were conducted in Stata 15 (StataCorp LP, College Station, TX, USA) and R version 0.98.932 (R Development Core Team, 2014).

## 3. Results

### 3.1. Population Data

The analysed population consisted of 3404 dogs (70.8%) registered as puppies in New Zealand and 1403 (29.2%) registered in an Australian state and imported to New Zealand either before or during their racing career. Male dogs accounted for 54.3% (*n* = 2609/4807) of the dogs that raced. Data from the four racing seasons (2013–2016) contained 175,322 eligible racing starts by 4807 individual dogs in 22,277 races. Overall, there was no change in the number of racing tracks (*n* = 7) used across the racing seasons, nor in the number of race meetings (Median: 449, inter-quartile range (IQR): 449–450), number of races (Median: 5586, IQR: 5525–5614), number of racing starts (Median: 43,923, IQR: 43,442–44,222), or number of individual greyhounds racing (Median: 2171, IQR: 2123–2205). 

The pattern of racing remained consistent across the years, with a median of 11,496 racing starts during winter (IQR: 11,311–11,631), a median of 10,853 racing starts during spring (IQR: 10,744–11,021), 10,582 starts during summer (IQR: 10,400–10,865) and 10,838 racing starts during autumn (IQR: 10,683–11,008). The number of races held remained consistent throughout winter (Median: 1457, IQR: 1435–1472), spring (Median: 1383, IQR: 1368–1402), summer (Median: 1354, IQR: 1327–1383) and autumn (Median: 1375, IQR: 1358–1395). 

Meetings were located at seven racetracks in seven regions across the North and South Islands of New Zealand. There were significantly more racing starts in the Canterbury (Median: 12,988, IQR: 12,945–13,102) and Whanganui (Median: 10,594, IQR: 10,310–10,846) regions each season (*p* < 0.05), where more than one race meeting was held each week, compared to the other five racing tracks (Auckland: Median: 5821, IQR: 5782–5853; Waikato: Median: 4320, IQR: 4219–4395; Manawatu: Median: 4843, IQR: 4769–4997; Otago: Median: 2682, IQR: 2671–2699; Southland: Median: 2499, IQR: 2415–2577).

The majority of races (89%; *n* = 19,835/22,277) had a full field of eight dogs, and races with seven or more runners accounted for 98% (*n* = 21,891/22,277) of all races. The age of the dogs, at the time of any racing start, ranged from 14 months through to 67 months with a median racing age of 31 months for New Zealand born dogs and 34 months for Australian born dogs. From the 175,322 racing starts, 65.8% (*n* = 115,518/175,322) of starts were dogs aged between 14 and 36 months; and a total of 99.99% (*n* = 175,312/175,322) of racing starts were completed by dogs aged between 14 and 72 months.

The distribution of races throughout the race grades, stratified by the age of the dog at the time of the race, is presented in Table 1. Dogs began their racing career at the earliest age of 14 months and 47% (*n* = 82,834) of races were run by dogs aged between 24 and 36 months. Maiden races were comprised primarily of dogs 14-month to 25-month old (64%), while the higher race classes (3, 4 and 5) were run by dogs predominantly older than 26 months of age (Table 1). 

The race distance ranged from 295 m to 779 m and races were categorised as sprint <457 m (*n* = 14,459, 65%), middle distance 457–599 m (*n* = 7404, 33%) or distance races ≥600 m (*n* = 414, 1.8%). The median age of the dogs racing varied by the race distance category (*p* < 0.001) (Sprint: Median: 32 months old, IQR: 26–40 months; Middle Distance: Median: 31 months old, IQR: 25–38 months; Distance: Median: 37 months old, IQR: 31–43 months). 

### 3.2. Cohort Data

A cohort of 2630 registered dogs had a racing career between 1/8/2013 and 31/7/2017. Of these, 1,718 dogs were born in New Zealand (65.3%) and 912 born in Australia (34.7%). In the cohort 53.9% of the dogs were male (*n* = 1417/2630) and 46.1% bitches (*n* = 1213/2630). Male dogs accounted for 60.6% (*n* = 553/912) of the Australian born dogs and 50.3% (*n* = 864/1718) of the New Zealand born dogs registered for racing. Of these 2630 dogs registered for racing, 2393 dogs (91%) had at least one race start and 237 dogs (9%) were registered but never raced. There were records for 2117 qualifying trials which included 404 (19.1%) Australian dogs and 1713 (80.9%) New Zealand dogs. Of the 2393 dogs that raced; 675 (28.2%) were Australian dogs and 1718 (71.8%) were New Zealand born dogs. The dogs that raced had a total of 97,973 starts in 21,571 races throughout the defined time period. Summary measures for the age dogs were registered for racing, completed their qualifying trial and completed their first racing start are presented in Table 2. There were significant differences between Australian and New Zealand dogs for the age they were registered, trialled and raced (*p* < 0.001) (Table 2). 

The median number of days between racing starts for each dog was 7 (IQR: 4–10). In 116 of 95,580 racing starts (0.12%), dogs raced on consecutive days and 5451 (5.70%) competed in races 2 days apart. Most racing was structured around a 7-day cycle (Figure 1), regardless of the different regions and racing seasons. The median did not change across different age groups based on the age of first racing start, nor did it change for the country where the dog was born. The autocorrelation function (ACF) plot (Figure 1b) represents the correlation in time between races where lag was set at 7 days. There was a sharp reduction in the correlation coefficient between the reference of 7 days (ACF = 1.00) and lag 1 (ACF = 0.085). The plot also shows decay in the correlation over a period of 7 weeks with some irregularity after this point.

The median number of starts per racing career was 35 (IQR: 16–59) with no significant difference in the number of racing starts between the Australian (Median: 33 starts, IQR: 16–55 starts) and New Zealand dogs (Median: 35 starts, IQR: 16–59 starts) (*p* = 0.15). The median racing career length was 424 days (IQR: 206–647 days). There was a significant difference between the career length in days for Australian (Median: 378 days, IQR: 183–583 days) and New Zealand (Median: 445 days, IQR: 216–672 days) dogs (*p* < 0.001). There was a significant difference between the length of career and the maximum racing grade reached during a dog’s racing career (*p* < 0.001). 

Regarding the number of days dogs had between races: 12.4% of Australian (*n* = 84/675) and 8.3% of New Zealand dogs (*n* = 143/1,718) had predominantly high-intensity racing intervals; 33.3% of Australian (*n* = 225/675) and 35.4% of New Zealand dogs (*n* = 143/1718) raced on a medium-intensity racing interval; 4.7% of Australian (*n* = 32/675) and 10.4% of New Zealand (*n* = 179/1718) dogs raced on a low-intensity racing interval; and 49.3% of Australian (*n* = 333/675) and 45.3% of New Zealand (*n* = 778/1718) dogs were not categorised due to having fewer than 50% of racing starts in each category. Dogs with a higher proportion of high frequency races had significantly more racing starts during their career compared with dogs that raced less frequently (*p* < 0.001). Within the New Zealand dogs, there was a significant difference between the frequency of racing category and the career length (*p* < 0.001), number of career racing starts (*p* < 0.001), and finish age (*p* < 0.001). There was a significant difference between the number of racing starts for the racing frequency groups in Australian dogs (*p* < 0.001), however, there was no difference between career length nor finish age. 

Summary measures for the age dogs finished their racing careers are presented in Table 2. There was a significant difference in finish age between New Zealand dogs (Median: 36 months, IQR: 29–43 months) and Australian dogs (Median: 39 months, IQR: 33–46 months) (*p* < 0.001). Dogs from Australia were significantly more likely to finish racing at a later age than New Zealand dogs (Log-rank *p*-value < 0.0001) (Figure 2).

However, there was no significant difference between the age of New Zealand and Australian born dogs at their last racing start when dogs that had achieved similar maximum racing class were compared (Table 3) (*p* > 0.2 for all groups).

## 4. Discussion

The objective of this study was to determine the pattern of racing and career duration for racing greyhounds in New Zealand. To the authors’ knowledge, this is the first attempt to describe trends in greyhound racing careers. The regularity of the greyhound racing industry, in terms of the scheduling and location of race meetings provide the opportunity for dogs to race on a weekly basis. The study population consisted of all the dogs that raced during four seasons and encapsulates information about the racing careers of these dogs. The number, frequency, type and distribution of races remains consistent across all racing seasons, indicating that this sample could be considered representative of the current racing population in New Zealand.

While information around the number of dogs that were born but never had a racing start was not available from the data examined in this study, previous reports from New Zealand have estimated that 20%–28% of dogs born were never registered for racing [18]. In the present study, of the dogs registered for racing, 9% failed to have a racing start in New Zealand. This figure is similar to the relatively low percentage loss of horses reported in the other two racing codes, and Tanner [12,13] demonstrated a 9% loss of Standardbred horses between registration and racing in New Zealand [13]. The moderately low number of dogs that did not race after registration suggests that there was genuine intention by owners and trainers to race their registered dogs.

The results from this study suggest that frequency of racing is driven by the regularity of race meetings at local racetracks. The pattern of racing is determined by the industry scheduling of race meetings as suggested by the limited variation in the number of days between racing starts, number of races and number of race meetings. Within each region, there was limited variation in the patterns of racing, suggesting that trainers regularly travel to their local racetracks. Exploring the relationship between patterns of racing and career duration identified dogs that typically raced with high-intensity racing intervals and an equal sized cohort that predominantly raced with low-intensity racing intervals. Dogs that primarily raced less than once a week had shorter career lengths and fewer racing starts during their career compared with dogs that raced more frequently. Although there are differences between inter-race intervals amongst individual dogs, the reason for and the impact of these differences, in terms of health and injuries, remains unknown. Further research exploring the racing patterns of dogs, more specifically, investigating the reason why racing patterns differ between dogs is required. The small number of Australian dogs in the low-intensity inter-racing interval category is potentially due to the quality of dogs being imported from Australia, as well as the older median age they commence racing. To maximize the potential return for these Australian dogs, they would be required to race frequently. Although this present study identified that there are differences between the inter-racing interval for individual dogs, more research is required to determine the reason dogs fall into each of these categories.

The frequency of racing (median 7 days between racing starts) is unique to the greyhound code, compared with the other racing codes where Thoroughbreds in New Zealand start in flat races a median of 5 times and jump races a median of 3 times each calendar year, and Standardbred pacers in New Zealand run a median of 7 times every calendar year while Standardbred trotters race a median of 8 times [19,20]. Concerns surrounding the frequency of racing in greyhounds falls back to whether the dogs are racing too frequently and whether this predisposes them to racing injuries. Previous reports have demonstrated a high percentage (10%–44%) of the injuries experienced by racing greyhounds involve central tarsal bone fractures [21,22,23]. The aetiology of the injury involves the accumulation of micro damage to distal limb bones through the application of cyclic stresses from training and racing [22,24,25]. The balance between bone fatigue, micro damage and the healing through adaptive remodelling is affected by the frequency of racing, as well as the amount of cyclic loading that is sustained during training and racing [23]. High accumulation of distance performed at racing speeds is seen to predispose Thoroughbred horses to musculoskeletal injuries as, in part, a result of skeletal and soft tissue damage from repeated loads of high speed or high intensity exercise [26]. No similar studies for racing greyhounds have been undertaken and thus there is a need to explore the impact that racing every seven days may have on the health and welfare of racing greyhounds.

Stevenson and colleagues [6] found an increase in the number of racing starts by greyhounds, over a period of two months was associated with decreased injury risk [6]. While this implies physical fitness is a protective measure, cumulative exposure of the greyhound’s musculoskeletal system to the stresses involved with racing and training, despite having not been well documented, are considered to increase the risk of cyclic overload [6,16,27]. In the current study, the high-intensity racing interval dogs have shorter careers with more racing starts compared with the low-intensity racing dogs. This enables high-intensity racing dogs to have more racing starts over a shorter racing career. The reasons for shorter racing careers and fewer racing starts throughout a career appear multifactorial. Reasons may include dog-related factors where dogs with poor performance ability qualify for fewer starts than those that succeed earlier on in their racing careers, or may be due to injuries that shorten the career length [16]. Since 2014, Greyhound Racing New Zealand have made a constructive effort to increase the opportunities to race in order to help extend the racing career length [28]. The number of racing starts by New Zealand greyhounds appears to be much greater than the data reported from Victoria, Australia. While Beer [16] did not specifically look at the whole career of racing greyhounds in Victoria, her work demonstrated that across a 6-year period, the median number of race starts for an individual dog was 10 (IQR: 4–25 starts). This current study reports a much higher number of racing starts with a median of 35 starts (IQR: 16–59).

The distribution of race age for Australian dogs suggests that these dogs are imported at different stages of their racing career. It is possible that the older median race age and first start age in Australian dogs compared with New Zealand bred dogs reported here are due to the imported dogs having previous racing starts in Australia, before commencing their racing career in New Zealand. This, in part, may explain the lower number of Australian dogs that trialled (*n* = 404) compared to raced (*n* = 675). Australian dogs that enter New Zealand with a previous suspension from racing are required to complete a satisfactory qualifying trial, whereas those that have been racing successfully in Australia before entering New Zealand are able to begin racing in the equivalent race grade they were competing in Australia. Australian dogs that have raced before entering the racing scene in New Zealand are imported at a later age and thus automatically have a shorter career duration and are older when they finish racing. Furthermore, the present study reported a greater number of Australian dogs than bitches. This, in part, could be due to the quality females being kept in Australia for breeding [16,29] as opposed to potential stud dogs needing to be of exceptional quality to remain in Australia for breeding. In addition, the age Australian dogs cease racing is older than New Zealand dogs. This could, in part, be due to the effort and expense associated with importing a racing greyhound. In the current study, proportionally more imported dogs reached the higher racing grades during their career, which demonstrates that quality dogs are being selected for import into the New Zealand racing scene.

Dockerty [30] and Helton [31] report mean peak race performance is reached at ages of 2 years to 2.4 years (24 to 29 months) respectively. While race age has an influence on peak race performance, experience also plays a role in performance and this develops over time [31]. In another study, peak performance in racing greyhounds in terms of race speed, occurred at a later age of 30–40 months [29]. The age distribution of racing greyhounds in New Zealand is similar to those reported for greyhounds racing in Ireland, where 50% of racing starts are by dogs of two years and under [29]. While speed information was not available for analysis in the current study, Dockerty [30] found that racing age is correlated with speed; where speed improves until greyhounds are approximately 25 months of age, and then falls at a steady rate. In the present study, career duration is limited by the age of the dog. Dogs of similar ability finished racing at a similar age regardless of the age the dog began racing, the country the dog was from, or the total number of racing starts. Increasing age, past the point of peak athletic performance, has been clearly linked to a decline in maximal strength and power, which in humans, is due to a multifactorial array of endocrine changes, nervous system changes and muscle atrophy amongst other factors [32]. There are limited statistics on the retirement age of canine athletes, however, this age has been crudely reported as 5–6 years for racing greyhounds [33] and 6 years for dogs in other sporting disciplines in an Australian report [34]. The results from this study report a much lower age of last race compared with previous work. 

The age at which dogs finish racing reflects a decline in peak athletic performance. Although there were differences between Australian and New Zealand dogs, greyhounds with similar racing ability, in terms of reaching the same maximum race grade, finished racing at the same age. Investigating the physiological responses to racing every 7 days, as well as quantification of training practices that occur between race days, is necessary in order to determine if patterns of training and racing, have an effect on career longevity of racing greyhounds.

## 5. Conclusions

The racing careers of greyhounds racing in New Zealand has been studied using a dataset containing details on all greyhound racing starts over four consecutive racing seasons (2013–2016). Data contained 175,322 eligible racing starts by 4807 individual dogs in 22,277 races. The number of racing tracks (*n* = 7), the number of race meetings, number of races, number of racing starts, and number of individual greyhounds racing showed no significant differences across the racing seasons. Furthermore, the pattern of racing and number of races remained consistent across the years, with more racing starts during winter compared to spring, summer and autumn. The majority of races (89%) had a full field of eight dogs, and races with seven or more runners accounted for 98% of all races. The age of the greyhounds racing ranged from 14 months through to 67 months with a median racing age of 31 months for New Zealand born dogs and 34 months for Australian born dogs. Dogs began their racing career at age 14 months, 47% of races were run as 2-year-olds (24 to 36 months). Maiden races were primarily run by dogs 14-month to 25-month old (64%), while the higher race classes (3, 4 and 5) were mostly run by dogs over 26 months of age.

This study has found that the pattern and frequency of racing is driven by the industry scheduling of race meetings. Dogs that primarily raced less than once a week had shorter career lengths and fewer racing starts during their career compared with dogs that raced more frequently. Differences between the inter-racing intervals for individual dogs are identified, but more research is required to determine the reason dogs fall into each of these categories. Furthermore, high-intensity racing interval dogs have shorter careers with more racing starts when compared with the low-intensity racing dogs. This enables high-intensity racing dogs to have more racing starts over a shorter racing career.

From our data we could not determine the impact that racing every seven days may be having on the health and welfare of racing greyhounds. But in their 2009 study Stevenson and colleagues found a decreased injury risk with an increase in the number of racing starts by greyhounds over a two-month period. It could be that physical fitness protects the greyhound’s musculoskeletal system to the stresses involved with racing and training, but it is likely that cumulative exposure to cyclic overload increase the risk of injury long-term. We believe there is a need to explore the impact that racing every seven days may have on the health and welfare of racing greyhounds and that will be the focus of future work.

The present study has found that dogs of similar ability finished racing at a similar age regardless of the age the dog began racing, the country it was from, or the total number of racing starts it had. The age at which dogs finish racing reflects a decline in peak athletic performance. Although there were differences between Australian and New Zealand dogs, greyhounds with similar racing ability, in terms of reaching the same maximum race grade, finish racing at the same age. 

## Figures and Tables

**Figure 1 animals-10-00796-f001:**
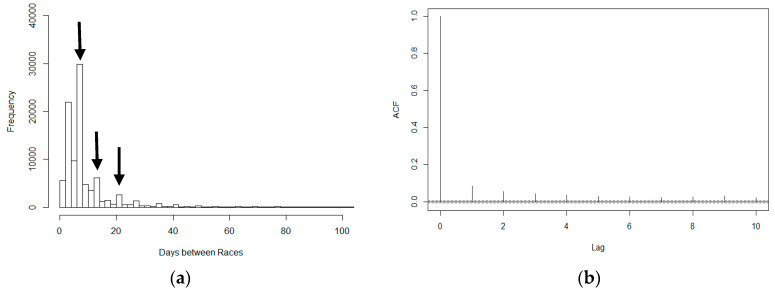
Frequency of racing, in terms of time between racing starts, in a cohort of 2393 racing greyhounds followed across 95,580 racing starts between the 2013 and 2016 seasons: (**a**) Frequency histogram of the number of days between consecutive races for individual greyhounds. The highest 1% of values have been removed. The arrows above the histogram demonstrate the peaks at 7 days, 14 days and 21 days respectively; (**b**) Autocorrelation function (ACF) showing the correlation in the raw residuals as a function of number of days between successive races for individual dogs. The dashed grey lines represent approximate 2-sided critical bounds for the autocorrelation at α = 0.01.

**Figure 2 animals-10-00796-f002:**
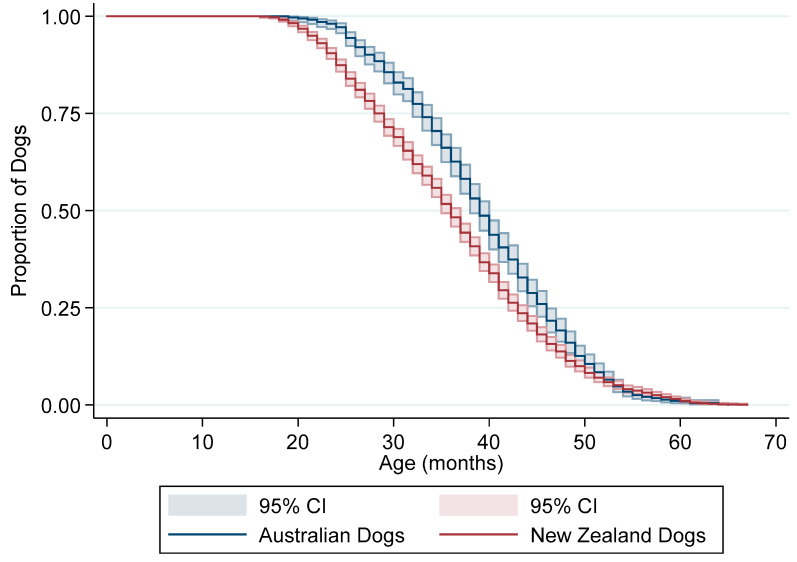
Kaplan–Meier “survival” curve of the age of final racing start (months) stratified by country for a cohort of racing greyhounds in New Zealand. The coloured area around each line represents 95% confidence intervals.

**Table 1 animals-10-00796-t001:** The distribution (number and percentage) of greyhounds racing by age and racing class for the 2013–2016 racing seasons in New Zealand.

Age (Month Category)	Race Grade							No. of Starters
	Class 0	Class 1	Class 2	Class 3	Class 4	Class 5	Other	
% dogs per grade								
14–25	64.1	27.8	15.7	11.4	7.6	5.8	23.5	43,428
26–31	22.3	26.3	24.8	25.2	22.7	21.5	32.0	43,455
32–38	9.7	21.7	25.9	29.2	32.3	35.6	23.2	42,087
39–79	3.9	24.2	33.6	34.3	37.4	37.1	21.3	46,352
No. of starters by grade								
	25,494	56,933	28,211	20,128	20,882	14,861	8813	
No. of races by grade								
	3235	7246	3575	2555	2653	1866	1147	
% races by grade								
	14.5	32.5	16.0	11.5	11.9	8.4	5.1	

**Table 2 animals-10-00796-t002:** Summary information for the age at which racing milestones were reached, as well as the time between registering, trialling and racing, for a cohort of New Zealand greyhounds that raced during the 2013–2016 racing seasons. Differences between dogs from the two countries were tested with a Kruskal–Wallis test. IQR: inter-quartile range.

	Australian Dogs	New Zealand Dogs	Total	*p*-value
Age registered ^1^ (months)				
Median	24	19	20	<0.001
IQR	20–29	17–21	17–23	
N	912	1718	2630	
Age of qualifying trial ^1^ (months)				
Median	20	20	20	0.613
IQR	18–23	18–20	18–23	
N	404	1713	2117	
Age of first racing start ^1^ (months)				
Median	25	20	21	<0.001
IQR	21–30	18–23	19–25	
N	675	1718	2393	
Days between registering and racing				
Median	22	35	31	<0.001
IQR	13–45	19–71	17–64	
N	678	1692	2370	
Days between qualifying trial and racing				
Median	13	12	12	0.308
IQR	7–28	7–23	7–24	
N	364	1693	2057	
Age of last race (months)				
Median	39	36	37	<0.001
IQR	33–46	29–43	30–44	
N	675	1718	2393	

^1^ The age racing milestones were reached in New Zealand.

**Table 3 animals-10-00796-t003:** Summary information for the age finished racing (months) by the maximum race grade reached during the dog’s career for New Zealand and Australian born dogs, for a cohort of greyhounds that raced in New Zealand during the 2013–2016 racing seasons.

Maximum Race Grade	Australian Dogs			New Zealand Dogs			Total		
	N	Median Age (Months)	IQR	N	Median Age (Months)	IQR	N	Median Age (Months)	IQR
C0	34	27	24–34	388	26	23–31	422	26	23–31
C1	110	32	27–37	434	33	28–38	544	33	28–38
C2	81	39	33–43	251	40	34–44	332	39	34–44
C3	109	39	33–47	197	41	36–48	306	40	35–48
C4	69	40	35–45	155	41	37–47	224	41	36–47
C5	272	43	38–48	293	43	37–48	565	43	37–48
Total	675	39	33–46	1718	36	28–43	2393	37	30–44

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
