# Peer review of "Patterns of Racing and Career Duration of Racing Greyhounds in New Zealand"

_animals, 2020, doi:10.3390/ani10050796_

Round 1
Reviewer 1 Report
Congratulations for this paper, it raises relevant points for discussion about the current use of animals in our society.
From an aimal welfare perspective, it would be interesting to investigate:
1) the reasons why a pretty significant percentage (up to 28%) of dogs were never registered for racing or failed to have a racing start in NZ
2) behavioural differences and related potential AW issues in high-intensity racing VS low-intensity racing dogs
3) final destination of dogs that were never registered for racing or failed to have a racing start (companionship, AAA/AAT, euthanasia) and ethical implications
Reviewer 2 Report
While all data and statistics appear to be done appropriately, I believe a statement of funding and possible bias should be written and appear at the beginning of this article. It was funded by the greyhound racing industry so it is not surprising that the findings a indicate frequent racing promotes fewer injuries. Furthermore there is no discussion about the opposition to racing either greyhounds or horses. What is the disposal process for the greyhounds at the end of their career? What happens when they are injured? Are their injuries treated.
Reviewer 3 Report
Overall, this is a well presented and detailed study on the patterns of racing and career duration of greyhounds in New Zealand. You describe some interesting findings such as a short lag between race starts, a higher number of race starts especially for high intensity races, and that dogs of similar ability ended their career at a similar age. Your findings will be of interest to industry, welfare organisations and scientists alike, as a baseline to examine the impact of racing patterns on the health and welfare of racing greyhounds.
From the introduction, I did not understand the aim/reason for comparison between those bred in NZ and those imported from Aus. This was much the focus of your results though your main findings and conclusions are generally more of interest. Perhaps a clear introduction as to why you decided to look at the differences in these populations is needed?
Otherwise, I would have liked to see more detail in your statistical reporting - some patterns you describe were significance tested and others were not. Where P-values are provided, please also include the effect size and confidence intervals too where applicable. Out of interest, more detail on why your multilevel model was not further analysed would be interesting - why did you not remove correlating variables, or use an interaction term?
Please find some specific comments below.
Line 36 – do you have a reference for greyhound racing having the lowest racing animals and annual turnover? (compared to horses)
Line 48. Little or no work in New Zealand? If there is some, its pertinent to describe those studies here. Also perhaps describe studies in other countries – e.g. https://doi.org/10.1111/j.1748-5827.1999.tb03117.x please add reference to this article as it is missing but very relevant
Methods – I’m wondering why your main variable of interest was the country of origin? What was your hypothesis for testing the differences between these populations? Perhaps this can be made clearer in your introduction?
Line 122- was this pattern significance tested?
Line 153 – where there any differences in age of dog in race speed categories?
Line 200 – was this pattern significance tested?
Results general – where you provide P values, please also provide effect size and CI’s where appropriate.
Line 209-209, this looks to either be a multivariate model or separate P values are needed?
Line 279 but see above study to include here
Line 318 – additional bracket after reference
Line 335 – revisit sentence structure
